# A cautionary tale about properly vetting datasets used in supervised learning predicting metabolic pathway involvement

Erik D. Huckvale[1]*, Hunter N. B. Moseley[1,2,3,4]*

1 Markey Cancer Center, University of Kentucky, Lexington, Kentucky, United States of America,
2 Superfund Research Center, University of Kentucky, Lexington, Kentucky, United States of America,
3 Department of Molecular and Cellular Biochemistry, University of Kentucky, Lexington, Kentucky, United States of America, 4 Institute for Biomedical Informatics, University of Kentucky, Lexington, Kentucky, United States of America

* edhu227@uky.edu (EDH); hunter.moseley@uky.edu (HNBM)

## Abstract

The mapping of metabolite-specific data to pathways within cellular metabolism is a major data analysis step needed for biochemical interpretation. A variety of machine learning approaches, particularly deep learning approaches, have been used to predict these metabolite-to-pathway mappings, utilizing a training dataset of known metabolite-to-pathway mappings. A few such training datasets have been derived from the Kyoto Encyclopedia of Genes and Genomes (KEGG). However, several prior published machine learning approaches utilized an erroneous KEGG-derived training dataset that used SMILES molecular representations strings (KEGG-SMILES dataset) and contained a sizable proportion (~26%) duplicate entries. The presence of so many duplicates taint the training and testing sets generated from k-fold cross-validation of the KEGG-SMILES dataset. Therefore, the k-fold cross-validation performance of the resulting machine learning models was grossly inflated by the erroneous presence of these duplicate entries. Here we describe and evaluate the KEGG-SMILES dataset so that others may avoid using it. We also identify the prior publications that utilized this erroneous KEGG-SMILES dataset so their machine learning results can be properly and critically evaluated. In addition, we demonstrate the reduction of model k-fold cross-validation (CV) performance after de-duplicating the KEGG-SMILES dataset. This is a cautionary tale about properly vetting prior published benchmark datasets before using them in machine learning approaches. We hope others will avoid similar mistakes.

## Introduction

(Cellular) metabolism is the collection of biochemical transformations, i.e. chemical reactions involving reactant and product metabolites, that take place in and around cells. Metabolism is often described as a large network of reactions, which are classically subdivided into sections often referred to as pathways. Pathway implies a sequence of chemical reactions, but is better described as an interconnected local network of chemical reactions often with an implied

**Data Availability Statement:** The dataset, all code, and all results are provided in the following Figshare item: https://doi.org/10.6084/m9.figshare.22661185.

**Funding:** This work has been supported by the National Science Foundation [NSF 2020026 to HNBM] and the National Institute of Environmental Health Sciences [P42ES007380].

**Competing interests:** The authors have declared that no competing interests exist.

directionality across the local network. The Kyoto Encyclopedia of Genes and Genomes (KEGG) is a major scientific repository with a relatively comprehensive (but not complete) description of cellular metabolism in terms of a metabolic network of known reactions and associated metabolites in separate REACTION and COMPOUND database tables, respectively [1] [2, 3]. In the BRITE database entry br08901 (https://www.genome.jp/brite/br08901), KEGG describes a set of 12 high-level metabolism "pathways" that cover a majority of their reactions and associated metabolites. These 12 KEGG pathways are listed in Table 1.

Several research groups have downloaded the entries in the KEGG COMPOUND database associated with these pathways for the purpose of developing machine learning methods for predicting metabolic pathway based on compound chemical graph representation.

The first of these machine learning applications was Hu et al. [4] which used a random forest (RF) model [5] to predict metabolic pathway involvement, training a model to assign compounds to one or more of eleven KEGG pathways (see the 'Included As A Label In The KEGG-SMILES Dataset' column in Table 1). The 12th label was not included in these publications, likely because it resulted in poor machine learning performance as compared to the other labels [6].

Later publications introduced deep neural networks in an attempt to improve model performance for this machine learning task, including Baranwal et al. [7] who reports higher model performance scores using a graph convolutional network (GCN) [8] combined with an RF as well as a GCN by itself. While the model used by Hu et al. [4] and the dataset used to train it are not provided in their publication or supplemental material (at least we could not find it), Baranwal et al. provides the code they used as well as the dataset they created, as seen in Table 2. This dataset is a text file, with each line containing the associated KEGG pathway labels used for supervised learning preceded by a simplified-molecular-input-line-entry-system (SMILES) [9] representation of the metabolic compound, and is available in their GitHub repository [10]. We will refer to this dataset as the KEGG-SMILES dataset (Table 1). See Fig 1 for a preview of the dataset's contents. Yang et al. [11] claims to further improve on the performance of the model proposed by Baranwal et al. [7] using an attention-based [12] graph network. While the authors don't appear to provide code or data, they evidently used the same dataset to train their model considering the number of instances is the same and they also

**Table 1. Pathway categories and their inclusion.**

| Pathway Category Name | Included As A Label In The KEGG-SMILES Dataset |
|---|---|
| Carbohydrate Metabolism | Yes |
| Energy Metabolism | Yes |
| Lipid Metabolism | Yes |
| Nucleotide Metabolism | Yes |
| Amino Acid Metabolism | Yes |
| Metabolism of Other Amino Acids | Yes |
| Glycan Biosynthesis and Metabolism | Yes |
| Metabolism of Cofactors and Vitamins | Yes |
| Metabolism of Terpenoids and Polyketides | Yes |
| Biosynthesis of Other Secondary Metabolites | Yes |
| Xenobiotics Biodegradation and Metabolism | Yes |
| Chemical structure transformation maps | **No** |

While the KEGG Brite hierarchy includes 12 pathway categories, past publications on this machine learning task only used 11 of the 12. This includes the 'KEGG-SMILES' dataset introduced by Baranwal et al.

**Table 2. Availability of code and data for past publications.**

| Model / Feature Set | Data available | Code available | Dataset Size | Publication Date |
|---|---|---|---|---|
| **Hu *et al*. RF** [4] | No | No | 3,137 | December 2011 |
| **Baranwal *et al*. GCN/RF** [7] | Yes [10] | Yes [10] | 6,669 | April 2020 |
| **Baranwal *et al*. GCN** [7] | Yes [10] | Yes [10] | 6,669 | April 2020 |
| **Yang *et al*. GAT** [11] | No | No | 6,669 | December 2020 |
| **Du *et al*. MLGL-MP** [13] | Yes [14] | Yes [14] | 6,648 | June 2022 |

While Hu et al. and Yang et al. do not appear to provide data nor code, Baranwal et al. and Du et al. provide both. Hu et al. used an initial dataset that totaled to 3,137 instances while later publications used the dataset originating with Baranwal et al.

describe it as containing SMILES data in their manuscript. Finally, Du et al. [13] presents the most recent machine learning models generated from this KEGG-SMILES dataset (as of October 3$^{rd}$ 2023), with the same text file available in their GitHub repository [14]. Their dataset size is slightly smaller than previous publications, because feature vectors could not be generated from all chemical compound structures in the KEGG-SMILES dataset, causing these unconvertable entries to be dropped.

As detailed in Table 3, Baranwal et al. [7] reports on the accuracy, precision, and recall of Hu's RF method [4] and we provide the F1 score of Hu's method by calculating it based on the provided precision and recall. Baranwal et al. [7] additionally provides the accuracy, precision, recall, and F1 score of their own RF-GCN combination method as well as their GCN-only method. Yang et al. [11] reports the same four metrics though they state, "the F1-score takes into account the accuracy and recall to measure our model" (i.e. they calculated their F1 score using accuracy and recall rather than precision and recall). However, for consistency, we provide the F1 score calculated from their provided precision and recall using the standard F1 score formula.

The results from these past publications suggest a machine learning task that originates with Hu et al. and is improved upon by later publications, specifically improving the performance of models trained on the KEGG-SMILES dataset provided by Baranwal et al. However, the validity of the results presented by Baranwal et al. [7, 10] as well as Yang et al. [11] and Du et al. [13, 14] are highly questionable, particularly following our discovery of large numbers of exact duplicate entries in the KEGG-SMILES dataset, representing over 26% of the total dataset.

## Materials and methods

We began our analysis by forking the GitHub repository provided by Du et al. [14] and running their scripts in order to reproduce their results, initially with minor changes to their code to account for package-dependency issues. The scripts provided in the original repository are designed to only process and train / test their graph neural network model on a single train-

```
CC(=O)C(=O)O          0,1,4,5,7,8,9,10
CC(=O)SCCNC(=O)CCNC(=O)C(O)C(C)(C)COP(=O)(O)OP(=O)(O)OCC1OC(n2cnc3c(N)ncnc32)C(O)C1OP(=O)(O)O          0,1,2,4,5,6,8,9,10
OCC1OC(O)C(O)C(O)C1O          0,9
```

**Fig 1. The first 3 lines of the KEGG-SMILES dataset.** The KEGG-SMILES dataset, as created by Baranwal et al, was a tab-separated text file with the first column containing the SMILES representation of each metabolite and the second column containing the numeric identifier (0 to 10 inclusive) of each pathway category, the category identifiers being comma-separated.

**Table 3. Reported model performance of past publications.**

| Model / Feature Set | Accuracy (%) | Precision (%) | Recall (%) | F1 |
|---|---|---|---|---|
| **Hu _et al._ RF** [4] | 94.64 | 77.97 | 67.83 | 0.7254 |
| **Baranwal _et al._ GCN/RF** [7] | 97.58 ± .12 | 83.69 ± .78 | 83.63 ± .68 | 0.8366 |
| **Baranwal _et al._ GCN** [7] | 97.61 ± .12 | 91.61 ± .52 | 92.50 ± .44 | 0.9205 |
| **Yang _et al._ GAT** [11] | 97.50 ± .06 | 93.04 ± .28 | 93.22 ± .16 | 0.9313 |
| **Du _et al._ MLGL-MP** [13] | 98.64±0.47 | 95.26±2.25 | 94.21±1.94 | 0.9473 |

If available in the corresponding publication, we report on the standard deviation of the model performance metric across CV folds, as indicated by the ± symbol.

test split, despite the manuscript reporting to have trained and tested the model across ten CV folds. Fortunately, the original repo did contain the ten CV folds as CSV files (i.e. a train.csv and test.csv for each fold), so we modified the scripts further in order to process and train/evaluate the model across all ten folds rather than only one. Since these methods are stochastic and the model training was not seeded, we were not able to exactly reproduce the published results. However, we were able to closely approximate the original results by running each of the ten folds ten times (to account for differences in random seed and the stochastic nature of model initialization and training) for a total of 100 runs. This enabled us to take the standard deviations of the performance scores across the 100 runs, which provides better estimates of the standard deviations than what was previously published.

In the training/model evaluation script of Du et al. [14], the model is trained via stochastic gradient decent over 200 epochs (batch size of 256), specifically using the Adam optimizer [15]. The graph neural network was trained with a dropout layer of probability 0.2. The ReLu activation function was used for all but the final activation, in which a LeakyReLu was used with a negative slope of 0.2. A learning rate of 0.0005 was used for the Adam optimizer along with betas of 0.9 and 0.999, epsilon of $10^{-8}$, and a weight decay of 0.

For every CV fold, the test set is evaluated in each epoch and the scores reported are those from the epoch where the model performed best on the test set. In other words, the test set is evaluated multiple times and the highest scores are chosen from the multiple evaluations.

After re-running their scripts and observing the performance scores of their model trained on the original KEGG-SMILES dataset, we wrote an additional script that removed exact duplicate entries to generate a de-duplicated dataset, i.e. keeping the first occurrence of a duplicated entry and removing any recurring duplicates (note that we define a duplicate as both the features being exactly the same and the labels being exactly the same). The purpose of the deduplication is to allow a valid evaluation of model performance. But this dataset deduplication analysis necessitated further minor modification of the original scripts provided by Du et al. [14] in order to optionally take in either the KEGG-SMILES dataset containing duplicates or the de-duplicated version. That enabled us to produce model performance results on both the original dataset and de-duplicated dataset.

Finally, we wrote additional scripts to collect statistics on both the original dataset and the de-duplicated version. From the results generated from these scripts, we derived summary statistics as well as results from statistical analyses.

All scripts were written in the Python programming language [16] and the model was created using the PyTorch Geometric library [17] for creating deep neural network models specialized to handle irregularly shaped data such as graphs. PyTorch geometric is built on top of the deep learning library known as PyTorch [18]. Statistical analyses were performed using methods from the SciPy package [19]. All code and data for replicating these findings along

with instructions on doing so can be found in a Figshare item as supplemental material (see below). We also include copies of the GitHub repositories of Baranwal et al. [10] and Du et al. [14] in this Figshare item. If running the model training script, we recommend using a graphics processing unit (GPU) of up to 12 gigabytes memory to significantly reduce the runtime. However, it can be run very slowly on the central processing unit (CPU) alone.

**Figshare DOI:** https://doi.org/10.6084/m9.figshare.22661185

## Results

Table 4 shows three examples of duplicate entries in the KEGG-SMILES dataset, including the SMILES representation (features), the numeric pathway labels, and the line number that they appear in the file provided in the Baranwal et al. GitHub repository [14]. The lower line number of each duplicate is the first occurrence followed by recurrences on later line numbers. Notice that not only is the SMILES representation identical but also the labels. This is the case for all duplicates.

Table 5 shows both the number of compounds that correspond to each pathway category as well as the proportion of the compounds in the dataset that have a given pathway label. Note the fractions do not add up to one, since some entries are associated with more than one pathway label. Thus, these entries are counted more than once with respect to pathway inclusion. Table 5 additionally lists the percentage of duplicate entries for each pathway category as well as the total percentage of duplicates in the entire original dataset. In the de-duplicated dataset, we kept the initial occurrence of an entry with duplicates and removed each repeating occurrence. The resulting de-duplicated dataset had *4,929* instances. As with the original dataset, Table 5 provides the number of compounds per pathway label as well as their fraction in the de-duplicated dataset.

Two things stand out in Table 5: 1) the percentage of duplicates in every pathway category is higher than that of the overall dataset and 2) the proportion of the dataset in each pathway category in the de-duplicated dataset is lower than every pathway category in the original dataset. We suspected there was a relation between the number of times a unique entry occurs in the original dataset (occurring once is a non-duplicate and occurring multiple times is a duplicate) and the number of pathway labels said entry has. Table 6 provides the counts of the unique entries (i.e. from the collection of unique SMILES representations, same as the de-duplicated dataset) with a certain number of occurrences in the original dataset and a certain number of pathway labels. Note that the counts do not add up to a total equal to the size of the original dataset but rather that of the de-duplicated dataset since only unique entries are counted.

Table 6 can be compressed along the rows into two categories: the top row representing the unique entries that only occur once and the remaining rows representing the unique entries that occurred more than once. The columns can be compressed into two categories with the

**Table 4. Examples of duplicate instances.**

| Line Number | SMILES | Labels |
|:---:|:---:|:---:|
| 468 | C#CC (= O)O | 0,5 |
| 2225 | C#CC (= O)O | 0,5 |
| 5774 | C1 = CC2OC2c2ccccc21 | 10 |
| 5775 | C1 = CC2OC2c2ccccc21 | 10 |
| 1845 | C1 = NCCCC1 | 4,9 |
| 4437 | C1 = NCCCC1 | 4,9 |

**Table 5. Dataset statistics for the original dataset compared to the de-duplicated dataset.**

| Label ID | Pathway Category | Number Of Compounds In Dataset (Original) | Fraction Of Dataset (Original) | Percentage Of Duplicates | Number Of Compounds In Dataset (De-duplicated) | Fraction Of Dataset (De-duplicated) |
|---|---|---|---|---|---|---|
| 0 | Carbohydrate metabolism | 1126 | 0.169 | 67.05 | 371 | 0.075 |
| 1 | Energy metabolism | 750 | 0.113 | 72.80 | 204 | 0.041 |
| 2 | Lipid metabolism | 1066 | 0.16 | 38.93 | 651 | 0.132 |
| 3 | Nucleotide metabolism | 342 | 0.051 | 49.12 | 174 | 0.035 |
| 4 | Amino acid metabolism | 1440 | 0.217 | 54.37 | 657 | 0.133 |
| 5 | Metabolism of other amino acids | 597 | 0.09 | 59.80 | 240 | 0.049 |
| 6 | Glycan biosynthesis and metabolism | 325 | 0.049 | 64.00 | 117 | 0.024 |
| 7 | Metabolism of cofactors and vitamins | 948 | 0.143 | 44.83 | 523 | 0.106 |
| 8 | Metabolism of terpenoids and polyketides | 1483 | 0.223 | 35.13 | 962 | 0.195 |
| 9 | Biosynthesis of other secondary metabolites | 1906 | 0.287 | 35.78 | 1224 | 0.248 |
| 10 | Xenobiotics biodegradation and metabolism | 1452 | 0.218 | 32.58 | 979 | 0.199 |
| N/A | Total Dataset | 6,648 | N/A | 25.86 | 4,929 | N/A |

The number of compounds in each pathway category in both the original and de-duplicated datasets along with the fraction of the dataset that each label occupies, followed by the total number of compounds in both datasets. Note that the label fractions will not add up to one since some entries have more than one label. For the original dataset, we provide the percentage of duplicate entries in each pathway category followed by the total percentage of duplicates in the entire dataset.

first column representing entries with just one pathway label and the remaining columns being those with multiple labels. The resulting contingency table, i.e. Table 7, shows a non-random dependent relationship between the number of occurrences versus the number of labels, with the top left to bottom right diagonal weighted in the 2x2 table. This is mirrored in Table 6, but not as obvious.

**Table 6. Counts Of unique entries according to number of occurrences and number of pathway labels.**

| Number Of Occurrences In The Dataset | Number Of Pathway Labels | | | | | | | | | | |
|---|---|---|---|---|---|---|---|---|---|---|---|
| | 1 | 2 | 3 | 4 | 5 | 6 | 7 | 8 | 9 | 10 | 11 |
| 1 | 3983 | 66 | 24 | 12 | 11 | 6 | 1 | - | 3 | - | 7 |
| 2 | 327 | 168 | 10 | 7 | 1 | 1 | 4 | 1 | 1 | 1 | 2 |
| 3 | 46 | 52 | 27 | 3 | 1 | 1 | 1 | - | 1 | - | 2 |
| 4 | 13 | 14 | 15 | 7 | 3 | 1 | 2 | - | - | - | - |
| 5 | 2 | 4 | 11 | 4 | - | 2 | - | 1 | - | - | - |
| 6 | 4 | 5 | 5 | 7 | 3 | 2 | - | - | - | - | - |
| 7 | 3 | 3 | 7 | 2 | - | 1 | - | - | - | - | - |
| 8 | 2 | - | - | 1 | 1 | 1 | - | 1 | - | - | - |
| 9 | - | - | - | 1 | - | 1 | - | - | 1 | - | 1 |
| 10 | - | 2 | - | 2 | - | 2 | - | - | 1 | - | - |
| >= 11 | - | 2 | - | 3 | 7 | 2 | 3 | 2 | 1 | - | 1 |

The number of occurrences in the dataset is the number of times a unique entry (unique SMILES representation and corresponding labels) appears in the original dataset. A few unique compounds appeared up to 30 times, though we compress occurrences greater than or equal to 11 into the final row to simplify the table. The counts in each cell are the number of metabolites that have the specified number of occurrences and number of labels.

**Table 7. Unique entry occurrence compared to label count.**

| Unique Entry Occurrence | Only One Label | Multiple Labels |
|---|---|---|
| Occurs Once | 3983 | 130 |
| Occurs More Than Once | 397 | 419 |

Table 8 quantifies the average number of labels and the percentage of entries with multiple labels in the (original) KEGG-SMILES dataset, a subset with no duplicates in the original KEGG-SMILES dataset (non-duplicates), and a subset (technically a sublist since it contains duplicates) with duplicates in the original KEGG-SMILES dataset (duplicates). Unlike the metrics from Tables 6 and 7, we include all the entries as they are in the original dataset, i.e. instead of unique entries being counted once, each recurrence of a duplicate is included as well. We see in Table 8 that the non-duplicate subset had a significantly smaller average number of labels and percentage of entries with multiple labels than the original dataset and the subset containing the duplicates had significantly higher values of these same metrics. The ratio of the average number of labels between duplicates and non-duplicates is over 2.5 and the ratio of the percentage of entries with multiple labels between duplicates and non-duplicates is over 20. These differences also explain why the pathway fractions in Table 5 were all lower in the deduplicated dataset vs the original dataset.

Table 9 provides results of statistical tests corresponding to Tables 6–8. The Chi Square test of the contingency table (Table 6) resulted in a p-value of 0. Since the compressed contingency table (Table 7) had two rows and two columns, we used a Fisher's Exact test resulting in a p-value of $8.38 \times 10^{-254}$. Additionally, we performed a Mann-Whitney U test comparing the number of labels of non-duplicate entries to the number of labels of duplicate entries (Table 8) resulting in a p-value of 0. While the effect sizes of the chi-squared (Cramer V = 0.2995) and Mann Whitney U (common language effect size = 0.1952) statistical tests are considered weak (small), the Fisher exact test effect size (phi coefficient = 0.5693) is considered strong (moderately high). But no matter which method you prefer, the chance that the entries were randomly duplicated is pragmatically zero.

Table 10 shows a significant reduction in model performance after removing duplicate entries from the dataset. In particular, the precision drops by over 13%, the recall by over 12% and the F1 score by over 11%. Note that the results we produced by re-running the scripts of

**Table 8. Label quantities of the non-duplicate entries and duplicate entries compared to the original dataset.**

| Subset | Average Number Of Labels | Percentage Of Entries With Multiple Labels | Size |
|---|---|---|---|
| Original | 1.72 | 26.32 | 6,648 |
| Non-duplicates | 1.08 | 3.16 | 4,113 |
| Duplicates | 2.76 | 63.90 | 2,535 |

The average number of labels and percentage of entries with multiple labels are provided for the original dataset and compared to the subsets of non-duplicate entries and duplicate entries. Note that the size of the non-duplicates plus that of the duplicates is equal to the size of the total dataset since they are directly derived from the original dataset without any overlap (an entry is either a duplicate or it isn't) and all occurrences of duplicates are counted (e.g. if a unique entry occurs 3 times, it's counted 3 times in the size). So the above are the values calculated from all entries as they are in the original dataset. We see the non-duplicate subset had much lower values than those of the original and the duplicates subset had much higher values. The difference is even more extreme when comparing the non-duplicates directly to the duplicates.

Table 9. Results of statistical tests.

| Test Data | Statistical Test | Test Statistic | p-value | Effect Size Method | Effect Size |
|---|---|---|---|---|---|
| Complete Contingency Table | Chi-Squared | 4422.49 | 0 | Cramer's V | 0.2995 |
| Compressed Contingency Table | Fisher's Exact | 32.34 | 8.38x10^-254 | Phi Coefficient | 0.5693 |
| Number Of Labels | Mann-Whitney U | 2035425.5 | 0 | Common Language Effect Size | 0.1952 |

The Chi Square statistic is the result of the contingency table of Table 6 and the Fisher's exact test statistic is the result of the two-by-two contingency table of Table 7 (compressed version of the initial contingency table). The Mann-Whitney U test statistic results from comparing the number of labels of the non-duplicate entries to that of the duplicate entries (Table 8). The corresponding p-values of each test is displayed, where values of 0 are a result of the precision limitations of the implemented method. In addition to the statistical significance, the effect size is reported along with each method.

Du et al. slightly differ from those reported in their manuscript [13] (Table 3) as seen in Table 10. This is a result of the stochastic nature of neural network initialization and training algorithms. However, they fall well within a single standard deviation for precision, recall, and F1 score. Accuracy is with one standard deviation provided by Du et al, but not within the new standard deviation we calculated. In fact, all of the standard deviations we estimated from 100 folds of results were lower than the standard deviations we believed were estimated from just 10-folds worth of performance. Now, the drop in model performance between original and de-duplicated datasets is quite dramatic and fall well outside 5 standard deviations on all four metrics. Therefore, we can confidently conclude it's a result of the de-duplication rather than stochastic variation. Also, notice the dramatic increase in standard deviations for these 4 performance metrics from the reanalyzed original dataset and the de-duplicated dataset, typically by double or more. This demonstrates a dramatic drop in the robustness of the model training.

## Discussion

Machine learning researchers such as Zhao et al. [20] and Allamanis [21] have touched on the issues in validity that arise when machine learning datasets contain exact duplicates. Their results complement ours (Table 10), showing that machine learning datasets containing duplicates inflate model performance compared to the de-duplicated counterparts. The bias introduced by entry duplicates represents a non-random sampling and can result in overestimating the performance when evaluating machine learning models, considering duplicates occur both in the training and testing sets across CV folds. More generally, the presence of exact duplicates in a dataset is a type of data leakage, where data used in training is leaked into the testing [22, 23]. Therefore, the results from Baranwal et al. [7, 10], Yang et al. [11], and Du et al. [13, 14] have inflated model performances. Based on our results in Table 10. These model performances are most likely inflated by roughly 10% in precision, recall, and F1 score. In the case of the MLGL-MP in particular, we can see in the training/evaluation script of the model that the

Table 10. Model performance per dataset.

| Dataset | Accuracy (%) | Precision (%) | Recall (%) | F1 Score (%) | Dataset size |
|---|---|---|---|---|---|
| As Reported In Du et al. Manuscript | 98.64±0.47 | 95.26±2.25 | 94.21±1.94 | 94.73 | 6,648 |
| Original Dataset | 98.21±0.16 | 94.07±1.00 | 94.54±0.62 | 94.30±0.52 | 6,648 |
| Deduplicated Dataset | 96.58±0.32 | 83.50±2.28 | 81.07±2.38 | 82.24±1.76 | 4,929 |

The means are followed by ± followed by the standard deviation.

test set is evaluated in each training epoch: https://github.com/dubingxue/MLGL-MP/blob/main/MLGL-MP-%28fold-1%29/Training.py. The scores reported are those from the epoch that evaluated best, therefore using the test set for model selection. While best practice for machine learning is to fully train the model and evaluate the test set only once per CV iteration, we retained the evaluation methods of Du et al. in order to make a maximally accurate comparison.

The standard deviation also increases significantly in the de-duplicated version of the KEGG-SMILES dataset (Table 10), which is expected since the presence of duplicates was artificially increasing performance. However, the standard deviation was calculated from the same 10 CV folds used in each run, which does not provide a reliable estimate of the performance variance since the training and test sets do not change. In fact, it grossly underestimates the real performance variance, since only the stochastic model training process is changing over the 10 iterations per CV fold. But to make an apples-to-apples comparison, we were forced to use the same protocol.

While the results of their publications have been shown to be invalid, Baranwal et al. and Du et al. follow many of the best practices in supporting computational scientific reproducibility by making the dataset, results, and code available to support their publications [24–26]. This practice made it possible to detect the presence of duplicates in the KEGG-SMILES dataset and verify the inflated model performance. Unfortunately, Hu et al. and Yang et al. did not follow these best practices and thus their results cannot be independently evaluated and verified.

We see in Table 5 that the percentage of duplicates in every pathway category is significantly higher than the percentage of duplicates in the entire dataset. We additionally see that the proportions of the pathway categories in the original dataset are all higher than those in the deduplicated dataset. These unexpected results can be explained by Tables 6–9. We observe a statistically significant difference in the number of labels that non-duplicate entries have versus that of duplicate entries. This suggests that the duplication of entries is not random with respect to their number of pathway labels. The moderately strong phi coefficient of 0.5693 (Table 9), being positive, indicates the directionality of this non-random relationship: the duplication of entries is correlated with a higher number of pathway labels. This explains why individual pathway categories in the original dataset had much higher duplicate percentages than that of the overall dataset and why the proportion of entries in every category was higher in the original dataset than that of the de-duplicated version. The proportion of entries in each pathway category also acts as the proportion of positives or the proportion of the entries that map to that label as compared to not mapping to it (negative). When looking at each category individually, we observe that this dataset suffers from a class imbalance problem [27] (in supervised machine learning classification tasks) i.e. there are many more negative entries than positive entries. With class imbalance, a model can attain decent accuracy by primarily predicting the majority class. However, the number of true-positive predictions decreases when the model develops this tendency after training, resulting in a decrease in precision and recall. With the non-random duplication of entries that map to more pathway labels, we observe that the fraction of positive entries increases for every pathway category. This likely explains why both the original and de-duplicated datasets performed well in accuracy, but the de-duplicated dataset dropped in performance with precision and recall and the resulting F1 score.

One may suppose that the de-duplicated version of the KEGG-SMILES dataset is sufficient for a new benchmark dataset. However, there are other considerations, one of which is reproducibility. The only description of the creation of the KEGG-SMILES dataset is, "A dataset of 6669 compounds belonging to one or more of these 11 constituent pathway classes was downloaded (February 2019) from the KEGG database." This is without any description of the exact

REST API operations used to download these compounds and pathway associations. KEGG provides MOL file [28] representations but not SMILES [9] representations of their compounds, suggesting further processing after downloading the raw MOL file data but without any description of this processing. The code used to download and perform this data processing are also absent from the GitHub repository [10] where only the text file itself is present. The apparent lack of reproducibility obfuscates the exact methodology for constructing the original KEGG-SMILES dataset and introduces the possibility of other potential issues with it beyond entry duplication. Similar to the issue of reproducibility is the ability to update the dataset. KEGG periodically modifies and adds to their databases, so the metabolite entries available in KEGG and their associated pathways continue to be added and updated. Providing code enables reconstruction of the most complete dataset. The lack of code for creating the KEGG-SMILES dataset prevents updating the dataset as KEGG is updated.

We propose the need for a new benchmark dataset for this machine learning task that satisfies the requirements of reproducibility, completeness, and validity. Huckvale et al. [6] provides such a new benchmark dataset that contains all the known metabolite entries in KEGG COMPOUND, as of July 23rd 2023, with associated pathway labels after careful filtering to remove inappropriate entries (duplicates, non-metabolites, and entries with fewer than 7 non-hydrogen atoms), a construction method to keep the dataset up-to-date, and a thorough description of the construction method and rationale. This new dataset includes 5,683 unique entries, far exceeding the 4,929 instances in the de-duplicated KEGG-SMILES dataset. Therefore, we recommend this new dataset and construction method over the duplicate-containing KEGG-SMILES dataset.

## Acknowledgments

We thank Dr. Robert Flight for feedback on the machine learning evaluation methodology utilized in this work. We thank the University of Kentucky Center for Computational Sciences and Information Technology Services Research Computing for their support and use of the Lipscomb Compute Cluster and associated research computing resources.

## Supplemental materials

Supplemental material which includes all data, results, and code used to generate the results is available in the Figshare item: https://doi.org/10.6084/m9.figshare.22661185

## Author Contributions

**Conceptualization:** Hunter N. B. Moseley.

**Data curation:** Erik D. Huckvale.

**Formal analysis:** Erik D. Huckvale.

**Funding acquisition:** Hunter N. B. Moseley.

**Methodology:** Hunter N. B. Moseley.

**Resources:** Hunter N. B. Moseley.

**Software:** Erik D. Huckvale.

**Supervision:** Hunter N. B. Moseley.

**Validation:** Erik D. Huckvale, Hunter N. B. Moseley.

**Visualization:** Erik D. Huckvale.

**Writing – original draft:** Erik D. Huckvale.

**Writing – review & editing:** Hunter N. B. Moseley.

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
