## [Decision Letter · Decision Letter 0]

14 Jan 2024

PONE-D-23-33052A cautionary tale about properly vetting datasets used in supervised learning predicting metabolic pathway involvementPLOS ONE

Dear Dr. Moseley,

Thank you for submitting your manuscript to PLOS ONE. After careful consideration, we feel that it has merit but does not fully meet PLOS ONE’s publication criteria as it currently stands. Therefore, we invite you to submit a revised version of the manuscript that addresses the points raised during the review process.

We look forward to receiving your revised manuscript.

Kind regards,

Manikkam Rajalakshmi

Academic Editor

PLOS ONE

“This work has been supported by the National Science Foundation [NSF 2020026 to HNBM]. “

“This work has been supported by the National Science Foundation [NSF 2020026 to HNBM].  We thank Dr. Robert Flight for feedback on the machine learning evaluation methodology utilized in this work. We thank the University of Kentucky Center for Computational Sciences and Information Technology Services Research Computing for their support and use of the Lipscomb Compute Cluster and associated research computing resources.”

“This work has been supported by the National Science Foundation [NSF 2020026 to HNBM]. “

Reviewers' comments:

Reviewer's Responses to Questions

**Comments to the Author**

1. Is the manuscript technically sound, and do the data support the conclusions?

Reviewer #1: Yes

Reviewer #2: Yes

2. Has the statistical analysis been performed appropriately and rigorously? 

Reviewer #1: Yes

Reviewer #2: Yes

3. Have the authors made all data underlying the findings in their manuscript fully available?

Reviewer #1: Yes

Reviewer #2: Yes

4. Is the manuscript presented in an intelligible fashion and written in standard English?

Reviewer #1: Yes

Reviewer #2: Yes

5. Review Comments to the Author

Reviewer #1: This article A cautionary tale about properly vetting datasets used in supervised learning predicting metabolic pathway involvement is structured properly authors did the statistical analysis. But some changes they have to do these are. 1.proof reqd the entire manuscript 2) Authors should draw graphical abstract 3) Compare your study am with existing systems

Reviewer #2: The manuscript entitled “A cautionary tale about properly vetting datasets used in supervised learning predicting metabolic pathway involvement” has many mistakes, authors need to rectify many portions.

Should there be more precision in explaining the machine learning applications, such as the specific methodologies used by Hu et al, Baranwal et al, Yang et al, and Du et al?

Could there be confirmation or additional details on the machine learning model used by Hu et al [4], especially the dataset they used and its characteristics?

Should there be more information on why the 12th label was excluded in the machine learning predictions, and what criteria were used for inclusion or exclusion?

Should there be confirmation or additional details on the availability of datasets, particularly the KEGG-SMILES dataset, and how researchers can access and utilize it?

Could there be more explanation on the format of the KEGG-SMILES dataset, such as the structure of each line and how the SMILES data is represented?

Should there be more clarification on how the dataset size varies across different publications, and what factors led to the variations?

Could there be more precision in reporting metrics, especially regarding the accuracy, precision, recall, and F1 score calculations?

Should there be an explanation or investigation into the reasons for the large numbers of exact duplicate entries in the KEGG-SMILES dataset, and how it might affect the validity of the results?

Could there be more information on how the validity of results presented by Baranwal et al [7], Yang et al [11], and Du et al [13] is being questioned, and what specific concerns ?

Should there be more details on how the scripts were modified to process and train/evaluate the model across all ten cross-validation folds, and how the original repository handled these folds?

Could there be an explanation for not being able to exactly reproduce the published results due to the lack of seeding in model training? How did running each of the ten folds multiple times contribute to approximating the original results?

Should there be more details on the training/model evaluation script, especially regarding the parameters used, the role of stochastic gradient descent, and the selection process for reporting scores?

Should there be more information on the modifications made to Du et al's original scripts to accommodate either the KEGG-SMILES dataset containing duplicates or the de-duplicated version?

Could there be more details on the statistics collected and the results derived from the scripts, especially the summary statistics and the nature of the statistical analyses performed?

Should there be clarification on why the fractions in Table 5, representing the proportion of compounds in the dataset with a given pathway label, do not add up to one? How does the possibility of entries being associated with more than one pathway label contribute to this?

Could there be an explanation of the rationale behind de-duplicating the dataset? How does removing recurring duplicates impact the subsequent analyses?

Should there be more interpretation of the two notable observations in Table 5, especially regarding the higher percentage of duplicates in every pathway category and the lower proportion of the dataset in each pathway category in the de-duplicated dataset?

Could there be more details on the suspected relation between the number of times a unique entry occurs in the original dataset and the number of pathway labels it has? How does Table 6 illustrate this relationship?

Should there be clarification on how Table 6 is compressed to create Table 7? What does the resulting contingency table reveal about the relationship between the number of occurrences and the number of labels?

Could there be more information on the significance of the metrics in Table 8, especially the average number of labels and the percentage of entries with multiple labels? How do these metrics differ between the non-duplicate subset, the original dataset, and the subset containing duplicates?

Should there be more context on the statistical tests performed in Table 9, such as the Chi Square test, Fisher's Exact test, and Mann-Whitney U test? How do these tests contribute to the overall interpretation of the results?

Could there be more interpretation of the effect sizes mentioned, such as Cramer V, common language effect size, and phi coefficient? How do these effect sizes contribute to understanding the strength of the observed relationships?

Should there be more discussion on the pragmatic implications of the findings, especially the assertion that the chance that the entries were randomly duplicated is pragmatically zero? What does this imply for the validity and reliability of the dataset?

Good Luck!

6. PLOS authors have the option to publish the peer review history of their article (what does this mean?). If published, this will include your full peer review and any attached files.

Reviewer #1: No

Reviewer #2: No

---

## [Author Response · Author response to Decision Letter 0]

26 Jan 2024

Response:

We have addressed all of these requirements.

Issue 1:

Response:

We believe the manuscript conforms to PLOS One’s style requirements.

Issue 2:

Response:

All data, code, and results are available in the following Figshare item: https://doi.org/10.6084/m9.figshare.22661185

Issue 3:

“This work has been supported by the National Science Foundation [NSF 2020026 to HNBM]. “

Response:

The funders had no role, so please use the following statement:

Issue 4:

“This work has been supported by the National Science Foundation [NSF 2020026 to HNBM].  We thank Dr. Robert Flight for feedback on the machine learning evaluation methodology utilized in this work. We thank the University of Kentucky Center for Computational Sciences and Information Technology Services Research Computing for their support and use of the Lipscomb Compute Cluster and associated research computing resources.”

“This work has been supported by the National Science Foundation [NSF 2020026 to HNBM]. “

Response:

We have removed the funding information from the Acknowledgement section.

Issue 5:

Response:

We have checked the references.

Reviewer #1: 

This article A cautionary tale about properly vetting datasets used in supervised learning predicting metabolic pathway involvement is structured properly authors did the statistical analysis. But some changes they have to do these are. 

Response:

We thank the reviewer for the positive comments. We have addressed each issue raised.

Issue 1:

1.proof reqd the entire manuscript 

Response:

We have proofread the entire manuscript. But if we have still missed anything, please let us know.

Issue 2:

2) Authors should draw graphical abstract 

Response:

It is quite hard to create a graphical abstract for a paper with no figures. But we have made a try at creating a graphical abstract that describes data leakage, which is the central problem we detected.

Issue 3:

3) Compare your study am with existing systems

Response:

With all due respect, there is nothing to compare to. We detected a catastrophic data leak in the results generated for 3 papers. The authors of the first paper have acknowledged the problem with a comment on our preprint and said they are publishing a correction to their flawed paper: 

Erik D. Huckvale and Hunter N.B. Moseley. "A cautionary tale about properly vetting datasets used in supervised learning predicting metabolic pathway involvement" bioRxiv 2023.10.03.560711 (2023). https://doi.org/10.1101/2023.10.03.560711

We have also published a sister paper demonstrating a properly created dataset and classification results using 3 different machine learning methods:

Erik D. Huckvale, Christian D. Powell, Huan Jin, and Hunter N.B. Moseley. "Benchmark dataset for training machine learning models to predict the pathway involvement of metabolites" Metabolites 13, 1120 (2023). https://doi.org/10.3390/metabo13111120

Reviewer #2: 

The manuscript entitled “A cautionary tale about properly vetting datasets used in supervised learning predicting metabolic pathway involvement” has many mistakes, authors need to rectify many portions.

Response:

Respectfully, we disagree. Most of the issues and questions raised by the reviewer are asking for more detail and clarification, but do not identify mistakes in our manuscript. We have addressed these issues by adding more detail and clarification, when warranted. The authors of Baranwal et al have already acknowledged the data leakage problem in the dataset they generated in a comment posted on our preprint:

Erik D. Huckvale and Hunter N.B. Moseley. "A cautionary tale about properly vetting datasets used in supervised learning predicting metabolic pathway involvement" bioRxiv 2023.10.03.560711 (2023). https://doi.org/10.1101/2023.10.03.560711

Issue 1:

Should there be more precision in explaining the machine learning applications, such as the specific methodologies used by Hu et al, Baranwal et al, Yang et al, and Du et al?

Response:

Our manuscript is focused on the catastrophic data leakage in the dataset and its effects on model evaluation. The specific methodologies used in the machine learning methods is secondary. However, our manuscript describes what these methods are:

“The first of these machine learning applications was Hu et al [4] which used a random forest (RF) model [5] to predict metabolic pathway involvement, training a model to assign compounds to one or more of eleven KEGG pathways (see the ‘Included As A Label In The KEGG-SMILES Dataset’ column in Table 1). The 12th label was not included in these publications, likely because it resulted in poor machine learning performance as compared to the other labels [6].

Later publications introduced deep neural networks in an attempt to improve model performance for this machine learning task, including Baranwal et al [7] who reports higher model performance scores using a graph convolutional network (GCN) [8] combined with an RF as well as a GCN by itself. While the model used by Hu et al [4] and the dataset used to train it are not provided in their publication or supplemental material (at least we could not find it), Baranwal et al provides the code they used as well as the dataset they created, as seen in Table 2. This dataset is a text file, with each line containing the associated KEGG pathway labels used for supervised learning preceded by a simplified-molecular-input-line-entry-system (SMILES) [9] representation of the metabolic compound, and is available in their GitHub repository [10]. We will refer to this dataset as the KEGG-SMILES dataset (Table 1). Yang et al [11] claims to further improve on the performance of the model proposed by Baranwal et al [7] using an attention-based [12] graph network.”

Issue 2:

Could there be confirmation or additional details on the machine learning model used by Hu et al [4], especially the dataset they used and its characteristics?

Response:

We are limited to what authors provided in Hu et al [4], which does NOT include their dataset, model, code, or full results. This is a problem in many older machine learning papers that did not follow best practices for supporting computational scientific reproducibility. Please refer to the following paper in Science on best practices in publishing computational results:

Stodden V, McNutt M, Bailey DH, Deelman E, Gil Y, Hanson B, Heroux MA, Ioannidis JP, Taufer M. Enhancing reproducibility for computational methods. Science. 2016 Dec 9;354(6317):1240-1.

Issue 3: 

Should there be more information on why the 12th label was excluded in the machine learning predictions, and what criteria were used for inclusion or exclusion?

Response:

We can only speculate on why the authors of the prior papers did not include the 12th pathway label. 

However, our sister paper highlights a likely reason since the 12th pathway label provided the poorest classification results:

Erik D. Huckvale, Christian D. Powell, Huan Jin, and Hunter N.B. Moseley. "Benchmark dataset for training machine learning models to predict the pathway involvement of metabolites" Metabolites 13, 1120 (2023). https://doi.org/10.3390/metabo13111120

The manuscript mentions this possibility:

“The 12th label was not included in these publications, likely because it resulted in poor machine learning performance as compared to the other labels [6].”

Issue 4: 

Should there be confirmation or additional details on the availability of datasets, particularly the KEGG-SMILES dataset, and how researchers can access and utilize it?

Response:

Baranwal et al and Du et al provided their datasets. This is clearly indicated in Table 2 of our manuscript. Also, we have included their datasets in our Figshare item: 

https://doi.org/10.6084/m9.figshare.22661185

Hu et al and Yang et al did NOT provide their datasets, code, models, or full results, and thus, did not follow best practices in support computational scientific reproducibility.

Issue 5: 

Could there be more explanation on the format of the KEGG-SMILES dataset, such as the structure of each line and how the SMILES data is represented?

Response:

The KEGG-SMILES dataset is a simple text file that includes a SMILES string followed by the list of pathway labels as simple numbers. We have added a figure (Figure 1) showing the first few entries in the dataset, along with a basic description of the format.

Issue 6: 

Should there be more clarification on how the dataset size varies across different publications, and what factors led to the variations?

Response:

The manuscript clearly present how the dataset varies across different publications in Table 2. The factors for the variations are mention in the manuscript:

“Yang et al [11] claims to further improve on the performance of the model proposed by Baranwal et al [7] using an attention-based [12] graph network. While the authors don’t appear to provide code or data, they evidently used the same dataset to train their model considering the number of instances is the same and they also describe it as containing SMILES data in their manuscript. Finally, Du et al [13] presents the most recent machine learning models generated from this KEGG-SMILES dataset, with the same text file available in their GitHub repository [14]. Their dataset size is slightly smaller than previous publications, because feature vectors could not be generated from all chemical compound structures in the KEGG-SMILES dataset, causing these unconvertable entries to be dropped.”

Issue 7:

Could there be more precision in reporting metrics, especially regarding the accuracy, precision, recall, and F1 score calculations?

Response:

We already include 4 digits of precision, which is likely higher precision than these metrics warrant. We also include standard deviations, which statistically characterizes the variation in these metrics. This is all presented in Tables 3 and 10.

Issue 8: 

Should there be an explanation or investigation into the reasons for the large numbers of exact duplicate entries in the KEGG-SMILES dataset, and how it might affect the validity of the results?

Response:

We have already identified a data leakage problem caused by the large number of exact duplicates, which the authors of Baranwal et al have verified exists. At the time we submitted this manuscript, we did not have access to the code that generated the dataset. Given our familiarity with using the KEGG API (please see our kegg_pull publication), we know that certain approaches for using this REST API will likely produce list of entries with duplicates. Also, the authors of Baranwal et al have made updates to their code repository and included a script that generates a dataset via the KEGG API. However upon inspection, this script does not create the original dataset in a tab-delimited kegg_classes.txt file. It creates a CSV file called keggdb2.csv.

Issue 9: 

Could there be more information on how the validity of results presented by Baranwal et al [7], Yang et al [11], and Du et al [13] is being questioned, and what specific concerns ?

Response:

It is a data leakage problem arising from a large number of duplicate entries in the dataset. The authors of Baranwal et al have confirmed the problem exists. A large part of the Discussion section is devoted to describing the problem:

“Machine learning researchers such as Zhao et al [20] and Allamanis [21] have touched on the issues in validity that arise when machine learning datasets contain exact duplicates. Their results complement ours (Table 10), showing that machine learning datasets containing duplicates inflate model performance compared to the de-duplicated counterparts. The bias introduced by entry duplicates represents a non-random sampling and can result in overestimating the performance when evaluating machine learning models, considering duplicates occur both in the training and testing sets across CV folds. More generally, the presence of exact duplicates in a dataset is a type of data leakage, where data used in training is leaked into the testing [22] [23]. Therefore, the results from Baranwal et al [10] [7], Yang et al [11], and Du et al [14] [13] have inflated model performances. Based on our results in Table 10. These model performances are most likely inflated by roughly 10% in precision, recall, and F1 score. In the case of the MLGL-MP in particular, we can see in the training/evaluation script of the model that the test set is evaluated in each training epoch: https://github.com/dubingxue/MLGL-MP/blob/main/MLGL-MP-%28fold-1%29/Training.py. The scores reported are those from the epoch that evaluated best, therefore using the test set for model selection. While best practice for machine learning is to fully train the model and evaluate the test set only once per CV iteration, we retained the evaluation methods of Du et al in order to make a maximally accurate comparison.

The standard deviation also increases significantly in the de-duplicated version of the KEGG-SMILES dataset (Table 10), which is expected since the presence of duplicates was artificially increasing performance. However, the standard deviation was calculated from the same 10 CV folds used in each run, which does not provide a reliable estimate of the performance variance since the training and test sets do not change. In fact, it grossly underestimates the real performance variance, 

---

## [Editor Report · Decision Letter 1]

13 Feb 2024

A cautionary tale about properly vetting datasets used in supervised learning predicting metabolic pathway involvement

PONE-D-23-33052R1

Dear Dr. Hunter N. B. Moseley,

We’re pleased to inform you that your manuscript has been judged scientifically suitable for publication and will be formally accepted for publication once it meets all outstanding technical requirements.

Kind regards,

Manikkam Rajalakshmi

Academic Editor

PLOS ONE